# Development and Validation of the “Lying Flat” Tendency Scale for the Youth

**DOI:** 10.3390/bs13110915

**Published:** 2023-11-09

**Authors:** Huanhua Lu, Jun Hou, Anqi Huang, Jinli Wang, Feng Kong

**Affiliations:** 1School of Marxism, China University of Geosciences (Beijing), Beijing 100083, China; huanhualu@cugb.edu.cn (H.L.);; 2School of Psychology, Shaanxi Normal University, Xi’an 710062, China

**Keywords:** “lying flat”, scale development, reliability, validity

## Abstract

In recent years, “lying flat” has been enthusiastically pursued by young people in China, and it is worth studying its cause and social impact. However, there is still a lack of measurement tools that can scientifically evaluate an individual’s tendency for “lying flat.” In this study, a 6-item “Lying Flat” Tendency Scale was developed and cross-validated for reliability and validity in different samples from China. The findings demonstrated that the scale showed good internal consistency in three different samples; both exploratory factor analysis and confirmatory factor analysis supported the single dimension model of the scale, indicating good construct validity; the LFTS total score was negatively correlated with the satisfaction of basic psychological needs, happiness index, and positive emotions, and was positively correlated with negative emotions; simultaneously, the LFTS total score was also significantly positively correlated with the choice of “lying flat” behavior in the simulated situation. These results show that the scale has good validity and reliability, and can be used as a measuring tool for subsequent empirical research. It will help to promote the development of empirical research on the phenomenon of “lying flat”, help to understand the causes and consequences of “lying flat” more deeply, and also help to find effective ways to help young people break out of the “lying flat” dilemma.

## 1. Introduction

In 2021, “lying flat” became the annual hot term on the Internet and was selected as one of the “top ten buzzwords of the year” announced by “Bite words” [1]. The term “lying flat” can be traced back to the fandom term “lying flat and let oneself be laughed at” which appeared in 2016 [1,2]. In the first half of 2021, a post titled “lying flat is justice” was published on Baidu posting bar, and the author recounted his “lying flat” philosophy of how to step back in a world full of competition and busyness, claiming that “lying flat” is the movement of the wise [1,2]. This post used “lying flat” to characterize the state of not working, not consuming, and not socializing, which gave “lying flat” a specific sociological meaning [1,2]. Subsequently, the term “lying flat” has been fermented and spread through online media, has rapidly gained popularity as a vehicle for emoticons and popular quotations, and has been enthusiastically pursued by the youth in China. As a result, “lying flat” has become another “life philosophy” that triggered the youth network carnival, and has also become one of the brightest labels of the youth subculture.

“Lying flat” refers to the state of some youths who choose to give up their efforts and passively escape when the pressure they are under breaks through the individual’s psychological threshold [2]. In fact, “lying flat” is not born out of nowhere; it normally follows the “Buddhist” culture and the “funeral” culture [3]. There are similar forms in other countries or regions, such as the British NEET, the American Boomerang Kids, the Japanese low desire group, and the Korean social animal spirit [4]. “Lying flat” is a specific cultural phenomenon produced by the intensification of social competition and rapid social development, which is a typical modernity problem [5]. Due to the fear of fierce social competition, some young people choose to actively withdraw from the competition, and attempt to retreat, escape, and give up in the way of fierce social competition, which is the active abandonment and retreat of a “good life” [3,6,7,8,9,10]. Some other young people have also tried hard to pursue their own ideals, but become repeatedly frustrated, thus their inner sense of helplessness continues to accumulate, and they finally choose to give up the struggle and naturally embark on the road of “lying flat” [3,6,7,8,9,10]. Whether it is the former, or the latter, they have shown the common characteristics of “lying flat”: no hope for life, no goals, no desire, no progress, no action, resting on the status quo, unwilling to struggle, do not want to work hard, do not want to move up, just getting by, negativity and laziness, confusion, and abandonment [3,6,7,8,9,10,11,12]. 

Why do some young people choose “lying flat” in the face of the high-pressure environment of “involution”? Self-determination theory emphasizes the degree of self-determination in human behavior, viewing motivation as a continuum from unmotivated, external to internal motivation [13,14]. Self-determination theory suggests that the social environment enhances internal motivation and facilitates the internalization of external motivation by satisfying an individual’s three basic psychological needs (BPN): autonomy, competence and connection, which promotes the individual’s behavior; conversely, when the social environment fails to satisfy an individual’s BPN, it weakens the individual’s internal motivation [13,14]. Previous studies have showed that when an individual’s BPN are met, he or she can be internally motivated to work, have higher job satisfaction [15], be more persistent in the face of difficulties [13,14], and be less likely to give up or be emotionally exhausted [16], and conversely, when an individual’s BPN are not met, he or she is less motivated to achieve [13,14,17]. Self-determination theory constructs the theoretical model of “environment—the satisfaction of BPN—work motivation—behavior” [13,18], according to which we can analyze the reasons for the formation of “lying flat” among young people. In the face of enormous social pressure, some young people’s BPN are not satisfied, which greatly reduces their achievement motivation, and finally, they choose “lying flat “ [19,20]. That is, the lower the degree of satisfaction of BPN, the higher the tendency for “lying flat.“

In the face of enormous pressure, some young people choose “lying flat” to relieve their inner anxiety, but can “lying flat” really make individuals happier? Song et al. (2022) argued that choosing “lying flat” during the best years of one’s life is not the best choice for youth, and it is not conducive to the realization of one’s self-worth, and ultimately will only make one’s quality of life worse and worse, as it is hard to feel safe, happy, and fulfilled in such a life [5]. Previous empirical studies have also shown that a sense of purpose and meaning can make an individual happier [13,14,21,22,23], busyness can make an individual happier [24,25], and prolonged inactivity will increase negative thoughts and make people unhappier [26,27]. Therefore, we can deduce that “lying flat” will lead to the loss of opportunities for personal growth, the lack of a sense of purpose, meaning and fulfillment, and ultimately to a lower and lower sense of well-being, more and more negative emotions, and fewer and fewer positive emotions.

Since “lying flat” became a buzzword on the Internet, it has caused extensive discussion in the theoretical world [1,2,3,5,6,7,8,9,10,11,19,20,28,29,30,31]. Many scholars believe that “lying flat” will have many negative effects on individuals and society: it will not only aggravate the value confusion of the youth, but also promote the bad trend of social “decadence”, and even affect the progress of the modernization process of the whole country [5,6,28,29,30,31]. Therefore, we must take the “lying flat” problem of the youth seriously, analyze in depth the reasons for its emergence and popularity, and further explore how to guide the youth out of the “lying flat” predicament. However, due to the lack of measuring tools, the phenomenon of “lying flat” only remains at the stage of theoretical research. Therefore, we do not know yet what percentage of young people choose “lying flat” in real life? What is the current situation of these “lying flat” young people? What are the key factors contributing to their choice of “lying flat”? How does “lying flat” affect their lives? What kind of coping strategies can really help them? In order to answer these questions, there is an urgent need for a measuring tool to conduct empirical research. The purpose of our study was to develop a self-report scale to assess the “lying flat” tendency for young people, so as to provide a measuring tool for subsequent empirical research. 

Our study contained three sub-studies. In Study 1, we developed a six-item “Lying Flat” Tendency Scale (LFTS) based on an open-ended questionnaire, expert opinions, and exploratory factor analysis (EFA). In order to assess the measurement properties of the LFTS, in Study 2, we evaluated the reliability and construct validity of the LFTS in another sample; additionally, we examined the relationship between the “lying flat” tendency and the satisfaction of BPN, positive and negative emotions, and happiness index to further evaluate the validity of the LFTS. In Study 3, we evaluated the predictive validity of the LFTS by examining the relationship between the “lying flat” tendency and the choice of “lying flat” behavior in the simulated situation in the third sample. In short, we attempted to develop and cross-validate the reliability and validity of the LFTS through three studies. 

## 2. Study 1

Study 1 aimed to develop a scale to measure the tendency for “lying flat” of the youth group, which will provide a scientifically valid measurement tool for the study of “lying flat.” In this study, we first combined the open-ended survey and literature research to determine the core features of “lying flat”, accordingly prepared the items of the LFTS, and formed the initial version of the scale based on expert opinions. Then the scale was revised based on the results of the preliminary test and expert opinions to form the formal version of the LFTS.

### 2.1. Method

#### 2.1.1. Open-Ended Survey 

First, we investigated the core characteristics of “lying flat” through an open-ended survey. Specific questions were as follows: (1) What are the behaviors of people who tend to “lying flat” in life? (2) Are you a “lying flat” person? If you are, how do you usually behave? 

The open-ended survey was administered to 270 subjects from China, all of whom were young people in the 18–25 age group. Since open-ended questionnaires took a long time for the subjects, 50 subjects dropped out and 220 questionnaires were received. Of these, 5 questionnaires were voided due to illegible handwriting and 8 questionnaires were voided due to the fact that what was written did not have any relation to the questions, thus, 207 valid questionnaires were obtained in the end.

The text information collected from the survey was organized and Table 1 shows the typical descriptions of “lying flat.”

Combining the results of the open-ended survey and the descriptions of the characteristics of “lying flat” in the previous literature (see Section 1), the core characteristics of “lying flat” were determined as follows: lack of clear life goals, no desire and no want; negative attitude towards life, resting on the status quo, not thinking of making progress; not working hard, not striving, just getting by, and choosing to run away from problems.

#### 2.1.2. Development of the “Lying Flat” Tendency Scale

Based on the core characteristics of “lying flat” summarized above, the items of the LFTS were developed. Each item was revised according to the expert opinions, and the initial scale with 6 items was formed. It was a 4-point scale, with 1 indicating “very inconsistent”, 2 indicating “a bit inconsistent”, 3 indicating “more consistent”, and 4 indicating “very consistent.” The higher the score, the higher the tendency for “lying flat.”

#### 2.1.3. Preliminary test of the “Lying Flat” Tendency Scale

##### Participants

A total of 138 valid questionnaires were randomly collected through an online platform. All subjects were young Chinese in the age group of 18–25 years: 58 males, 80 females.

##### Measures

The initial version of the LFTS compiled above.

The Marlowe Crowne Social Desirability Scale (MCSD): It was the most widely used instrument for measuring social desirability, with higher total scores indicating greater social desirability [32]. 

### 2.2. Results

#### 2.2.1. Exploratory Factor Analysis 

KMO and Bartlett’s spherical tests were performed. The results showed that the KMO coefficient was 0.875, indicating suitability for factor analysis. And the Bartlett’s spherical test (*χ^2^* = 545.713, *df* = 15, *p* < 0.001) indicated the possibility of shared factors within the variables, which satisfied the prerequisites for factor analysis.

Factor analysis and item selection were conducted. Common factors were extracted using principal component analysis and orthogonal rotated maximum variance method, and the number of factors was determined according to the following criteria: (1) eigenvalues > 1; (2) the factor loading of the item was greater than 0.4; and (3) the items loaded heavily on only one factor. The results showed that six items were included under one common factor, and the factor loading of each item was greater than 0.4, with a total explanation rate of 69.212% (Table 2).

#### 2.2.2. Internal Consistency Reliability

The results showed that the overall internal consistency coefficient (Cronbach’s α coefficient) of the scale was 0.910, indicating that the reliability of the scale was good.

#### 2.2.3. Development of the Formal Version of the “Lying Flat” Tendency Scale

In order to further optimize the scale, some of the subjects were interviewed after the survey with the question: “Do you have any suggestions for this scale?” Some subjects indicated that some words in the scale (e.g., “laziness”, “lying flat” etc.) were formulated too negatively. In addition, a correlation analysis was conducted between each item score of the LFTS and the MCSD total score, and the results showed that item 3 (“I am a lazy person”) was significantly correlated with social desirability (*r* = 0.178, *p* = 0.036). Therefore, the formulation of each item was revised according to expert opinions. Finally, the formal version of the LFTS was developed (Table 3).

## 3. Study 2

In Study 1, we developed the LFTS, but needed to further evaluate its measurement properties. In Study 2, we evaluated the validity and reliability of the LFTS in a new sample. We used confirmatory factor analysis (CFA) to evaluate the construct validity of the LFTS, and examined the correlation between LFTS scores and the satisfaction of BPN, happiness index, and positive and negative emotions to further evaluate the validity of the LFTS.

### 3.1. Method

#### 3.1.1. Participants

We adopted a random sampling method and collected a total of 607 questionnaires. All subjects were young Chinese in the age group of 18–25 years: 374 males, 233 females.

#### 3.1.2. Measures

LFTS: The scale contained 6 items and was scored on a 4-point scale. The total score of the scale was used to represent the subject’s tendency for “lying flat”, and the higher the score, the higher the tendency for “lying flat.”

Positive Affect and Negative Affect Scale: There were nine positive emotions and nine negative emotions, with scores ranging from 1 (very little or none) to 5 (very much), and the higher the score, the more intense the individual’s emotional experience [33]. The Cronbach’s α coefficient for the positive emotion dimension and the negative emotion dimension were 0.917 and 0.839 in this study, respectively.

Happiness Index Scale: This scale contained two parts: the overall affective index and life satisfaction. The former contained 8 items that described the emotional content from different perspectives, while the latter contained only 1 item that examined the level of satisfaction with life in general. The total score was calculated by adding the average score of the overall affective index to the score of life satisfaction. The higher the score, the stronger the happiness index (Cronbach’s α = 0.824 in this study). 

Basic Psychological Need Satisfaction Scale (BPNS): This scale had 21 items [13,34]. Seven items measured autonomy needs (e.g., I feel I can make autonomous decisions about how to live my life); six items measured connection needs (e.g., I do like the people I hang out with); and eight items measured competence needs (e.g., I have recently become capable of learning new knowledge or skills) [13,34]. High scores indicated a high level of BPN satisfaction [13,34] (Cronbach’s α = 0.877 in this study). 

### 3.2. Results

#### 3.2.1. Descriptive Statistical Analysis of All Variables

Table 4 presents the descriptive statistics for each variable. The results showed that each scale score conformed to a normal distribution, with the skewness and kurtosis of each scale score lying between −1 and 1. The independent samples *t*-test showed that the gender difference in the positive emotion scores was borderline significant, with males (M = 33.00, SD = 7.05) having higher positive emotion scores than females (M = 31.90, SD = 6.36), while the gender differences in the scores of the remaining variables (including “lying flat” tendency, satisfaction of BPN, happiness index, and negative emotion) were not significant.

#### 3.2.2. Common Method Bias Test

The Harman’s Single-Factor Test was used to test for common method bias, which has been used by many studies in the past [35,36]. And it was found that the explanation rate of the first factor was 24.45% (less than 40%), indicating that the common method bias was not serious. Currently, more and more researchers are using the comprehensive CFA marker technique by Williams et al. (2010) to test common method bias, which is the most advanced method available [37,38,39,40,41]. Unfortunately, our study did not measure a marker variable that was theoretically unrelated to substantive variables and could not use the CFA marker technique.

#### 3.2.3. Internal Consistency Reliability

The internal consistency test of the LFTS was conducted, and the results showed that the Cronbach’s α coefficient of the scale was 0.819, indicating that the reliability of the scale was good.

#### 3.2.4. Confirmatory Factor Analysis 

A confirmatory factor analysis (CFA) of the LFTS was conducted based on a one-factor model (Figure 1). The results showed that the one-factor model had acceptable fit indices (*χ^2^/df* = 4.039, *RMSEA* = 0.071, *CFI* = 0.974, *TLI* = 0.957, *IFI* = 0.974, *NFI* = 0.966), indicating that the scale had acceptable construct validity.

#### 3.2.5. Validity Analysis 

Correlation analysis showed that the LFTS total score was significantly negatively correlated with the satisfaction of BPN (*r* = −0.361, *p* < 0.001), happiness index (*r* = −0.389, *p* < 0.001), and positive emotions (*r* = −0.166, *p* < 0.001), and was significantly positively correlated with negative emotions (*r* = 0.182, *p* < 0.001). 

In addition, correlation analysis between the CFA-obtained factor scores of “lying flat” and the additional variables (age, gender, positive emotion, negative emotion, well-being index, the satisfaction of BPN) was conducted. The results showed that CFA-obtained factor scores of “lying flat” were also significantly negatively correlated with the satisfaction of BPN (*r* = −0.359, *p* < 0.001, Table 5), happiness index (*r* = −0.390, *p* < 0.001, Table 5), and positive emotions (*r* = −0.160, *p* < 0.001, Table 5), and significantly positively correlated with negative emotions (*r* = 0.177, *p* < 0.001, Table 5).

## 4. Study 3

Although we examined the reliability and validity of the LFTS in Study 2, it was not clear whether LFTS scores could predict individuals’ “lying flat” behavior. Therefore, in Study 3, we evaluated the predictive validity of the LFTS by correlating LFTS scores with the choice of “lying flat” behavior in the simulated situation in a new sample. 

### 4.1. Method

#### 4.1.1. Participants

A total of 317 valid questionnaires were randomly collected by combining online and offline methods, including 169 Chinese males and 148 Chinese females, with 146 persons aged 18–25 years, 122 persons aged 26–30 years, and 49 persons aged 31–35 years.

#### 4.1.2. Measures

LFTS: The scale was the same as that used in Study 2.

Scenario simulation tests: Subjects were presented with two scenarios and asked to select the behavior from the two behavioral options given (A for “lying flat” behavior, B for non-”lying flat” behavior). Each behavioral option was marked on a 5-point scale, with the higher score given by the subject representing their greater tendency to choose the behavior. The total score for choosing behavior A in the two scenarios was calculated as the subject’s willingness to choose the “lying flat” behavior in the simulated situation. In scenario 1, we set up a situation where final exams were approaching and students were studying hard to prepare for them. We asked the subjects to imagine if they were in such a situation, and if they would choose to cope with it in a casual way and only pursue a score of 60 points (“lying flat” behavior), or would they choose to prepare hard for the exam and strive for a high score (non-”lying flat” behavior)? In scenario 2, we set up an internship situation in a company. Half of the interns will be retained as full-time employees based on their performance during the internship. All interns worked hard and often worked overtime to exceed their workloads. We asked subjects to imagine if they were in such a situation, and if they would choose to give up the competition and get by (“lying flat” behavior), or would they choose to work hard and try to become a full-time employee (non-”lying flat” behavior)?

### 4.2. Results

#### 4.2.1. Internal Consistency Reliability

The Cronbach’s α coefficient for the scale was 0.785.

#### 4.2.2. Confirmatory Factor Analysis

CFA showed that the one-way model fit index was acceptable (*χ^2^/df* = 2.563, *RMSEA* = 0.070, *CFI* = 0.976, *TLI* = 0.948, *IFI* = 0.976, *NFI* = 0.961), proving that the construct validity of the scale was acceptable.

#### 4.2.3. Predictive Validity

The correlation between the LFTS total score and the total scores of option A (“lying flat” behavior) in the two scenarios was found to be significantly positive (*r* = 0.478, *p* < 0.001), indicating that the LFTS has good predictive validity.

## 5. Discussion 

The aim of this study was to develop a self-report scale to assess the “lying flat” tendency for the youth. In Study 1, we developed a short version of the LFTS, and EFA indicated that the scale had only one dimension. In Study 2, we evaluated the reliability and validity of the scale in a new sample. CFA showed that the one-factor model was acceptable, while significant negative correlations existed between the LFTS scores and the satisfaction of BPN, positive emotions, and happiness index, providing further evidence of good validity of the scale. Study 3 found that the LFTS scores significantly and positively predicted an individual’s “lying flat” behavior in simulated situations in the third sample, indicating good predictive validity of the scale. Additionally, internal consistency analyses in all three samples indicated that the scale had good reliability.

Based on the literature research, open-ended surveys, and expert opinions, our study determined the core characteristics of “lying flat” and developed a short version of the LFTS. EFA showed that the scale had only one dimension, and CFA in another sample supported the single-dimensional model. These results indicated that the scale had good construct validity.

In the face of an “involutional” social environment, why do some young people choose “lying flat”? As mentioned in the Introduction, self-determination theory emphasizes the importance of satisfying BPN for individual achievement motivation [13,14]. A large body of previous research highlighted the importance of satisfying BPN in promoting individual intrinsic motivation, performance, and well-being [13,14,15,16,17,18,42,43,44,45,46,47]. When BPN were fulfilled, individuals were more likely to be motivated, engaged, and satisfied in their actions; conversely, when BPN were frustrated or unmet, individuals may be less motivated and may exhibit behaviors such as disengagement and abandonment [13,14,15,16,17,18,42,43,44,45,46,47]. At present, in the face of ever-increasing social pressure, some young people are constantly frustrated in real life, and their BPN, such as sense of competence, autonomy, and connection are not satisfied. They lose confidence in the future and feel that they are unable to cope with the great social competition, and thus choose “lying flat” [19,20]. Therefore, based on the self-determination theory, it is not difficult to infer that unsatisfied BPN may be one of the reasons why individuals choose “lying flat.” Our study found that the lower the degree of satisfaction of BPN, the higher the tendency for “lying flat”, which was consistent with the theoretical hypothesis, and this confirmed that the validity of the LFTS was good. 

Since “lying flat” became a buzzword on the Internet, lots of scholars have discussed the after-effects of “lying flat” [5,6,28,29,30,31]. Many theoretical studies have argued that “lying flat” may have a negative impact on the individual’s sense of well-being and emotions: in the face of realistic difficulties, individuals with a higher tendency for “lying flat” constantly choose to escape and compromise, and eventually will only make their quality of life worse and worse, their happiness lower and lower, their negative emotions more and more, and their positive emotions less and less [5,6,28,29,30,31]. Indeed, many studies have explored the relationship between NEET status and mental health, with findings suggesting that compared to non-NEETs, NEET populations experience more stress and anxiety, lower feelings of self-worth and self-esteem, an increased risk of substance abuse, and depression [48,49,50,51,52]. Here, we used empirical studies to prove that individuals with a higher tendency for “lying flat” have a lower happiness index, fewer positive emotions, and more negative emotions, which was consistent with scholars’ opinions and previous findings on NEET, and further confirmed the validity of the LFTS.

Our study also found that the LFTS total score was significantly and positively correlated with the choice of “lying flat” behavior in the simulated situation. That is, the higher the tendency for “lying flat”, the more likely an individual was to choose “lying flat” behavior in the simulated situation. This result proved that the LFTS in our study had good predictive validity.

In addition, the internal consistency of the LFTS was found to be good in all of the three different samples in our study, which indicated that the scale had good reliability. Of course, future studies can further examine the test–retest reliability of the scale to evaluate the reliability of the scale more comprehensively.

As a hot social phenomenon, “lying flat” has received great attention from scholars. But these researches only stay in the stage of theoretical discussion [1,2,3,5,6,7,8,9,10,11,19,20,28,29,30,31], and there is an urgent need to carry out empirical research to better understand the causes and social impacts of “lying flat”, so as to take better countermeasures. The LFTS developed in our study had good reliability and validity, so it can be used as a measuring tool for the subsequent empirical research.

However, there were also some limitations. First, the samples in our study were mainly drawn from the youth group, and the applicability of the scale needs to be tested in a more diverse population in the future. Second, based on previous literature in the field of sociology, open-ended surveys, and expert opinions, our study determined the core characteristics of “lying flat”, accordingly developed the LFTS, and revised the scale based on EFA and experts’ opinions, which is in strict compliance with the requirements of psychometrics, but due to the lack of authoritative definitions and theories of “lying flat” in the field of psychology, the stability of the scale’s theoretical construct still needs to be further validated in future studies. Third, the advanced CFA marker technique by Williams et al. (2010) was not used to test for common variance bias in Study 2 due to the lack of measurements of a marker variable unrelated to the substantive variables, which is currently the optimal method to test for common method bias and is being used in more and more studies [37,38,39,40,41]. Finally, the present study only examined the correlation between the tendency for “lying flat” and the satisfaction of BPN, the happiness index, and positive and negative emotions using cross-sectional studies, and we need to further investigate the causal relationship between these variables in order to better understand the causes and consequences of “lying flat”.

In our study, the short version of the one-dimensional LFTS was developed in strict accordance with psychometric standardized scale development procedures, and the validity and reliability of the scale were evaluated in a series of studies with different samples. The results showed that the LFTS had good validity and reliability and could be used as a measuring tool for subsequent empirical studies. It will help to promote the development of empirical research on the phenomenon of “lying flat”, help to understand the causes and consequences of “lying flat” more deeply, and also help to find effective ways to help young people break out of the “lying flat” dilemma.

## Figures and Tables

**Figure 1 behavsci-13-00915-f001:**
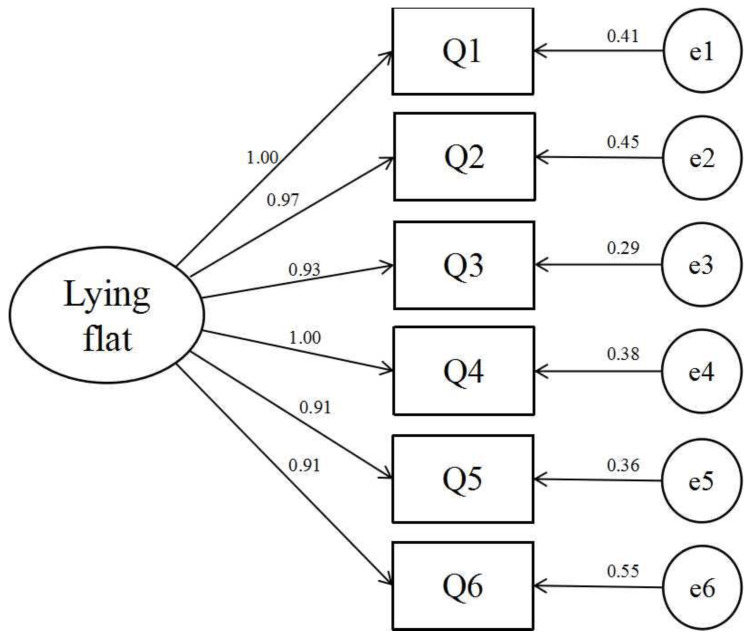
The one-factor CFA model of the “Lying Flat” Tendency Scale.

**Table 1 behavsci-13-00915-t001:** Typical descriptions of “lying flat”.

1. Not to make any effort and choose to give up when in trouble. 2. Having negative thoughts about study and life, having low expectations of oneself and the future. 3. Do nothing, just complain about themselves but not try to do anything or persist in their efforts. 4. No aggressiveness, only the pursuit of a comfortable life. 5. Unwilling to work hard because of the great pressure brought by fierce competition. 6. Do not know what their ideals and goals are, no desire and no want. 7. Negative attitude in the world, not thinking of making progress. 8. Loss of motivation and escape from reality.

**Table 2 behavsci-13-00915-t002:** EFA of the “Lying Flat” Tendency Scale.

Item	Factor Loading
Q1 I do not have any goals and pursuits for life and study, and I am really “lying flat” in action.	0.729
Q2 I feel that it is hard to change anything with my personal efforts, so I choose to give up the struggle.	0.820
Q3 I am a lazy person.	0.880
Q4 I am always reluctant to do those things that require effort to accomplish.	0.860
Q5 I often finish what I should do perfunctorily.	0.879
Q6 I am satisfied with my current state and don’t want to get involved in any competition to prove myself.	0.813

**Table 3 behavsci-13-00915-t003:** Formal version of the “Lying Flat” Tendency Scale.

Item
Q1 I do not have any goals and pursuits for life and study.
Q2 I feel that it is hard to change anything with my personal efforts, so I choose to give up the struggle.
Q3 I don’t work hard at anything.
Q4 I am always reluctant to do those things that require effort to accomplish.
Q5 I often finish what I should do perfunctorily.
Q6 I am satisfied with my current state and don’t want to get involved in any competition to prove myself.

**Table 4 behavsci-13-00915-t004:** Descriptive statistics of all variables.

Variables	M	SD	Skewness	Kurtosis
“lying flat” tendency	11.05	3.67	0.83	0.35
satisfaction of BPN	98.60	17.08	0.28	−0.08
happiness index	9.71	2.53	−0.12	−0.11
positive emotion	32.58	6.80	−0.14	−0.47
negative emotion	20.85	7.87	0.79	0.69

**Table 5 behavsci-13-00915-t005:** Correlations between CFA-obtained factor scores of “lying flat” and other variables.

	1	2	3	4	5	6	7
1. CFA-obtained factor scores of “lying flat”	—						
2. Satisfaction of BPN	−0.359 ***	—					
3. Happiness index	−0.390 ***	0.539 ***	—				
4. Positive emotion	−0.160 ***	0.587 ***	0.421 ***	—			
5. Negative emotion	0.177 ***	−0.439 ***	−0.314 ***	−0.231 ***	—		
6. Gender	0.009	0.049	−0.05	−0.079	0.056	—	
7. Age	−0.059	−0.027	−0.01	−0.072	0.005	0.024	—

Notes: *** *p* < 0.001, BPN = basic psychological needs, CFA = confirmatory factor analysis.

## Data Availability

The data used in this study are available on request.

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
