# Peer review of "Development and Validation of the “Lying Flat” Tendency Scale for the Youth"

_behavsci, 2023, doi:10.3390/bs13110915_

Round 1
Reviewer 1 Report
Comments and Suggestions for Authors
The study is indeed of high quality and required a tremendous amount of work and time of the authors. This effort has resulted in an excellent contribution to existing science and will certainly inspire other scientific teams to further investigate this new social-psychological phenomenon.
In an effort to help authors move their study to the level of scientific excellence, only 2 small inspirations can be recommended:
- Incorporate into the study more detailed results evaluating key items Positive Affect and Negative Affect Scale, Happiness Index Scale and BPN.
- Incorporate a little more scientific sources into the study (the current number of 30 sources is still sufficient, but a higher number of sources would further strengthen the relevance of the entire study and the developed unique tool).
There are some small formal imperfections, e.g., double spaces between sentences (line 33, 35, 37, etc.); short dashes instead of long ones (line 67); citation Song et al. it is not mentioned under the citation number directly in the text (line 73/74); small initial letter in Table 1 in item 6.
Good luck.
Author Response
We thank the reviewer for the helpful and detailed comments. In response to the issues raised, we have revised the manuscript accordingly. Please find the detailed responses in the attached document.

Reviewer 2 Report
Comments and Suggestions for Authors
The paper presents interesting research regarding a theme that is still much to be studied. As presented, it seems that this research is the first approach to the “lying flat” behaviour, particularly by Generation Z. As the authors stated, there is also a need to pursue further research to reach other targets since this behaviour can also be induced by burnout and other psychological and mental diagnosis.
Specifically regarding the paper presented to review, some improvements are suggested here:
Abstract – must be improved in terms of how it presents the research and specifically states the three studies separately. Since it is abstract, there is no need to do this, and an overall resume of the research is more adequate, particularly because it is necessary to reserve some sentences to describe the importance of the results and their contribution to science and society.
The paper structure and the methodologies used are adequate, but the materials and methods section can be better structured in a way that makes it easy to understand the importance of the different study phases and the interconnection between them. In my opinion, it would be better if this section would explain all three study phases and, after the result section, describe their results together in the three different phases.
The paper needs a conclusion section to explain the importance of the findings and their contribution to science and society.
Author Response

(The authors gave the same response as above.)

Reviewer 3 Report
Comments and Suggestions for Authors
Dear Author(s):
I had the opportunity to review your paper entitled "Development and Validation of the "Lying Flat" Tendency Scale for the Youth" submitted to the Journal of Behavioral Sciences. In this quantitative study, you gathered survey data from predominantly young “Chinese” (I guess – please indicate the geographical origin of participants) people aged 18-25 and developed and cross-validated a new scale for “lying flat tendency” using standard Structural Equation Modeling methods. Overall, your study is a nice read and well-crafted. Still, there is room for improvement. Please understand my review as a friendly hint on what you should pay most attention to before resubmitting a revision. Hopefully, my remarks will help you to improve your paper’s overall quality. Best of luck!
1. First, your theoretical positioning could be further improved. What is missing is a big why. Why do we need your study? Why should we care about young people in China partaking in such “lying flat” behavior. You mentioned negative consequences in the discussion section. Please add more reasons to motivate your study in the introduction. Insufficient research on a topic or controversies or disagreements may not be strong-enough reasons to motivate a study. What would happen if we did not answer your research question? What do we know already about the research question as it applies to theory and practice? And what do we not yet know, but urgently must learn? How does your study fill a significant gap in theory (and practice), or how does it address a fundamental anomaly (a big inconsistency between theory and reality)? What are your key contributions? You should aim for easy-to-digest takeaways in the study’s conclusions supported by the data. So please make the study more intriguing and captivating.
2. Theoretically, you mention self-determination theory in line 59 without any further explanations of basic assumptions and foundations of this theoretical lens. Please expand on why this theoretical lens fits your research goal best.
3. Methodologically, you lack some information. How and when did you collect the data of Study 1? 30% of invalid questionnaires (207 of 270) seems very high; why is that?
4. The item analysis in Table 2 makes no sense since – of course – there will be mean differences on the item level if you construct multiple groups based on these very items’ characteristics, namely top 27% vs. bottom 27% on their aggregate. I am not sure what you are trying to demonstrate but this proves absolutely nothing except for consistently low vs. high scores. Did you truly believe any of these items would produce insignificant mean differences? Rather, you should closely follow recommendations of classic test theory on developing reflective scales. By the way, aggregates follow the formative/composite logic instead of reflective mode. I would recommend to fully delete this confusing section 2.2.1. Moreover, on the item level, typically there is no discriminant analysis, only on the construct level via Fornell-Larcker ratios in covariance-based SEM or HTMT (heterotrait-monotrait) ratios in variance-based SEM. Item purification is based on low factor loading and high cross-loadings in the EFA. Since you only have one dimension, you should add the other latent measures in the EFA (social desirability, pos/neg emotions, well-being, etc.). Thereby, your EFA will be more informative due to a multi-dimensional space.
5. More critically, when it comes to subjectively perceived data, constant culturally induced method variance (CMV) typically experienced with highly collectivistic Eastern cultures (e.g., conformity bias, social desirability) will not vanish. Please use state-of-the-art CMV testing via the comprehensive CFA marker technique by Williams et al. (2010). This tool is very powerful by providing explicit diagnostics where common method variance occurs, on the measurement or structural level, or even both.
6. Why did you allow a residual covariance only between Q3 and Q4 in Fig. 1. This suggests cherry-picking and affects the lowest factor loading of Q3. Please do not manipulate results. Unless this is not a growth model, correlated residuals are not allowed and will invalidate the findings.
7. Why did you address a limitation of focusing only on 18-25 aged people? Study 2 reports diverse ages up to 31-35. Also, you should not say model fit is “good” (l. 256) since RMSEA >.05 but <.08 is only acceptable. Same for Cronbach’s alpha, <.8 is considered low, <.7 is considered unreliable, so .785 is not good but acceptable only. I would like to see a correlation table of the CFA-obtained factor scores of laying flat (instead of aggregates!) and the additional variables social desirability, age, gender, emotions, well-being, etc. This will inform us about the psychometric characteristics and potential differences between sexes and older participants who probably have more experience and therefore less tendencies of lying flat.
8. Minor issues: Please check some minor issues, e.g. line 74: add a space in “al.(2022).” Moreover, qualitative research is also empirical research, so “but” in line 90 does not apply. You probably mean conceptual or theoretical research instead of qualitative, or quantitative instead of empirical.
9. Your discussion is solid. However, I cannot identify truly novel ideas not already discussed in past literature. You could connect better with past findings and contribute to a consensus on this matter.
In conclusion, your paper requires more punch to be worthy of publication. I hope my assessment is not discouraging but helpful to further develop your theoretical thoughts and improve your methodological skills. All the best!
References
Williams, L. J., Hartman, N., & Cavazotte, F. (2010). Method variance and marker variables: A review and comprehensive CFA marker technique. Organizational research methods, 13(3), 477-514.
Comments on the Quality of English Language
I recommend a professional copy-editor.
Author Response
We thank the reviewer for the helpful and detailed comments. In response to the concerns raised we have revised the manuscript accordingly. Please find the detailed responses in the attached document.

Round 2
Reviewer 3 Report
Comments and Suggestions for Authors
Thank you for implementing most of my recommended actions. The manuscript improved significantly. Well done!
Comments on the Quality of English Language
Please run a final proofread.